# Sparsifying Networks via Subdifferential Inclusion

## Abstract

Sparsifying deep neural networks is of paramount interest in many areas, especially when those networks have to be implemented on low-memory devices. In this article, we propose a new formulation of the problem of generating sparse weights for a neural network. By leveraging the properties of standard nonlinear activation functions, we show that the problem is equivalent to an approximate subdifferential inclusion problem. The accuracy of the approximation controls the sparsity. We show that the proposed approach is valid for a broad class of activation functions (ReLU, sigmoid, softmax). We propose an iterative optimization algorithm to induce sparsity whose convergence is guaranteed. Because of the algorithm flexibility, the sparsity can be ensured from partial training data in a minibatch manner. To demonstrate the effectiveness of our method, we perform experiments on various networks in different applicative contexts: image classification, speech recognition, natural language processing, and time-series forecasting.

## 1 Introduction

Deep neural networks have evolved to the state-of-the-art techniques in a wide array of applications: computer vision (Simonyan & Zisserman, 2015; He et al., 2016; Huang et al., 2017), automatic speech recognition (Hannun et al., 2014; Dong et al., 2018; Li et al., 2019; Watanabe et al., 2018; Hayashi et al., 2019; Inaguma et al., 2020), natural language processing (Turc et al., 2019; Radford et al., 2019; Dai et al., 2019b; Brown et al., 2020), and time series forecasting (Oreshkin et al., 2020). While their performance in various applications has matched and often exceeded human capabilities, neural networks may remain difficult to apply in real-world scenarios. Deep neural networks leverage the power of Graphical Processing Units (GPUs), which are power-hungry. Using GPUs to make billions of predictions per day, thus comes with a substantial energy cost. In addition, despite their quite fast response time, deep neural networks are not yet suitable for most real-time applications where memory-limited low-cost architectures need to be used. For all those reasons, compression and efficiency have become a topic of high interest in the deep learning community.

Sparsity in DNNs has been an active research topic generating numerous approaches. DNNs achieving the state-of-the-art in a given problem usually have a large number of layers with non-uniform parameter distribution across layers. Most sparsification methods are based on a global approach, which may result in a sub-optimal compression for a reduced accuracy. This may occur because layers with a smaller number of parameters may remain dense, although they may contribute more in terms of computational complexity (e.g., for convolutional layers). Some methods, also known as magnitude pruning, use a hard or soft-thresholding to remove less significant parameters. Soft thresholding techniques achieve a good sparsity-accuracy trade-off at the cost of additional parameters and increased computation time during training. Searching for a hardware efficient network is another area that has been proven quite useful, but it requires a huge amount of computational resources. Convex optimization techniques such as those used in (Aghasi et al., 2017) often rely upon fixed point iterations that make use of the proximity operator (Moreau, 1962). The related concepts are fundamental for tackling nonlinear problems and have recently come into play in the analysis of neural networks (Combettes & Pesquet, 2020a) and nonlinear systems (Combettes & Woodstock, 2020).

This paper shows that the properties of nonlinear activation functions can be utilized to identify highly sparse subnetworks. We show that the sparsification of a network can be formulated as an approximate

subdifferential inclusion problem. We provide an iterative algorithm called subdifferential inclusion for sparsity (SIS) that uses partial training data to identify a sparse subnetwork while maintaining good accuracy. SIS makes even small-parameter layers sparse, resulting in models with significantly lower inference FLOPs than the baselines. For example, SIS for 90% sparse MobileNetV3 on ImageNet-1K achieves 66.07% top-1 accuracy with 33% fewer inference FLOPs than its dense counterpart and thus provides better results than the state-of-the-art method RigL. For non-convolutional networks like Transformer-XL trained on WikiText-103, SIS is able to achieve 70% sparsity while maintaining 21.1 perplexity score. We evaluate our approach across four domains and show that our compressed networks can achieve competitive accuracy for potential use on commodity hardware and edge devices.

## 2 RELATED WORK

### 2.1 INDUCING SPARSITY POST TRAINING

Methods inducing sparsity after a dense network is trained involve several pruning and fine-tuning cycles till desired sparsity and accuracy are reached (Mozer & Smolensky, 1989; LeCun et al., 1990; Hassibi et al., 1993; Han et al., 2015; Molchanov et al., 2017; Guo et al., 2016; Park et al., 2020). (Renda et al., 2020) proposed weight rewinding technique instead of vanilla fine-tuning post-pruning. Net-Trim algorithm (Aghasi et al., 2017) removes connections at each layer of a trained network by convex programming. The proposed method works for networks using rectified linear units (ReLUs). Lowering rank of parameter tensors (Jaderberg et al., 2014; vahid et al., 2020; Lu et al., 2016), removing channels, filters and inducing group sparsity (Wen et al., 2016; Li et al., 2017; Luo et al., 2017; Gordon et al., 2018; Yu et al., 2019; Liebenwein et al., 2020) are some methods that take network structure into account. All these methods rely on pruning and fine-tuning cycle(s) often from full training data.

### 2.2 INDUCING SPARSITY DURING TRAINING

Another popular approach has been to induce sparsity during training. This can be achieved by modifying the loss function to consider sparsity as part of the optimization (Chauvin, 1989; Carreira-Perpiñán & Idelbayev, 2018; Ullrich et al., 2017; Neklyudov et al., 2017). Dynamically pruning during training (Zhu & Gupta, 2018; Bellec et al., 2018; Mocanu et al., 2018; Dai et al., 2019a; Lin et al., 2020b) by observing network flow. (Mostafa & Wang, 2019; Dettmers & Zettlemoyer, 2020; Evci et al., 2020) computes weight magnitude and reallocates weights at every step. Bayesian priors (Louizos et al., 2017), $L_0$, $L_1$ regularization (Louizos et al., 2018), and variational dropout (Molchanov et al., 2017) get accuracy comparable to (Zhu & Gupta, 2018) but at a cost of $2\times$ memory and $4\times$ computations during training. (Liu et al., 2019; Savarese et al., 2020; Kusupati et al., 2020; Lee, 2019; Xiao et al., 2019; Azarian et al., 2020) have proposed learnable sparsity methods through training of the sparse masks and weights simultaneously with minimal heuristics. Although these methods are cheaper than pruning after training, they need at least the same computational effort as training a dense network to find a sparse sub-network. This makes them expensive when compressing big networks where the number of parameters ranges from hundreds of millions to billions (Dai et al., 2019b; Li et al., 2019; Brown et al., 2020).

### 2.3 TRAINING SPARSELY INITIALIZED NETWORKS

(Frankle & Carbin, 2019) showed that it is possible to find sparse sub-networks that, when trained from scratch, were able to match or even outperform their dense counterparts. (Lee et al., 2019) presented SNIP, a method to estimate, at initialization, the importance that each weight could have later during training. In (Lee et al., 2020) the authors perform a theoretical study of pruning at initialization from a signal propagation perspective, focusing on the initialization scheme. Recently, (Wang et al., 2020) proposed GraSP, a different method based on the gradient norm after pruning, and showed a significant improvement for moderate levels of sparsity. (Ye et al., 2020) starts with a small subnetwork and progressively grow it to a subnetwork that is as accurate as its dense counterpart. (Tanaka et al., 2020) proposes SynFlow that avoids flow collapse of a pruned network during training. (Jorge et al., 2020) proposed FORCE, an iterative pruning method that progressively removes a small number of weights. This method is able to achieve extreme sparsity at little accuracy expense. These

methods are not usable for big pre-trained networks and are expensive as multiple training rounds are required for different sparse models depending on deployment scenarios (computing devices).

### 2.4 EFFICIENT NEURAL ARCHITECTURE SEARCH

Hardware-aware NAS methods (Zoph et al., 2018; Real et al., 2019; Cai et al., 2018; Wu et al., 2019; Tan et al., 2019; Cai et al., 2019; Howard et al., 2019) directly incorporate the hardware feedback into efficient neural architecture search. (Cai et al., 2020) proposes to learn a single network composing of a large number of subnetworks from which a hardware aware subnetwork can be extracted in linear time. (Lin et al., 2020a) proposes a similar approach wherein they identify subnetworks that can be run efficiently on microcontrollers (MCUs).

Our proposed algorithm applies to possibly large pre-trained networks. In contrast with methods presented in Section 2.1, ours can use a small amount of training data during pruning and fewer epochs during fine-tuning. As we will see in the next section, a key feature of our approach is that it is based on a fine analysis of the mathematical properties of activation functions, so allowing the use of powerful convex optimization tools that offer sound convergence guarantees.

## 3 PROPOSED METHOD

### 3.1 VARIATIONAL PRINCIPLES

A basic neural network layer can be described by the relation:

$$y = R(Wx + b) \tag{1}$$

where $x \in \mathbb{R}^M$ is the input, $y \in \mathbb{R}^N$ the output, $W \in \mathbb{R}^{N \times M}$ is the weight matrix, $b \in \mathbb{R}^N$ the bias vector, and $R$ is a nonlinear activation operator from $\mathbb{R}^N$ to $\mathbb{R}^N$. A key observation is that most of the activation operators currently used in neural networks are proximity operators of convex functions (Combettes & Pesquet, 2020a;b). We will therefore assume that there exists a proper lower-semicontinuous convex function $f$ from $\mathbb{R}^N$ to $\mathbb{R} \cup \{+\infty\}$ such that $R = \text{prox}_f$. We recall that $f$ is a proper lower-semicontinuous convex function if the area overs its graph, its epigraph $\{(y, \xi) \in \mathbb{R}^N \times \mathbb{R} \mid f(y) \leqslant \xi\}$, is a nonempty closed convex set. For such a function the proximity operator of $f$ at $z \in \mathbb{R}^N$ (Moreau, 1962) is the unique point defined as

$$\text{prox}_f(z) = \underset{p \in \mathbb{R}^N}{\text{argmin}} \ \frac{1}{2}\|z - p\|^2 + f(p). \tag{2}$$

It follows from standard subdifferential calculus that Eq. (1) can be re-expressed as the following inclusion relation:

$$Wx + b - y \in \partial f(y), \tag{3}$$

where $\partial f(y)$ is the Moreau subdifferential of $f$ at $y$ defined as

$$\partial f(y) = \left\{ t \in \mathbb{R}^N \mid (\forall z \in \mathbb{R}^N) f(z) \geqslant f(y) + \langle t \mid z - y \rangle \right\}. \tag{4}$$

The subdifferential constitutes a useful extension of the notion of differential, which is applicable to nonsmooth functions. The set $\partial f(y)$ is closed and convex and, if $y$ satisfies Eq. (1), it is nonempty. The distance to this set of a point $z \in \mathbb{R}^N$ is given by

$$d_{\partial f(y)}(z) = \underset{t \in \partial f(y)}{\inf} \|z - t\|. \tag{5}$$

We thus see that the subdifferential inclusion in Eq. (3) is also equivalent to

$$d_{\partial f(y)}(Wx + b - y) = 0. \tag{6}$$

Therefore, a suitable accuracy measure for approximated values of the layer parameters $(W, b)$ is $d_{\partial f(y)}(Wx + b - y)$.

### 3.2 Optimization problem

Compressing a network consists of a sparsification of its parameters while keeping a satisfactory accuracy. Assume that, for a given layer, a training sequence of input/output pairs is available which results from a forward pass performed on the original network for some input dataset of length $K$. The training sequence is split in $J$ minibatches of size $T$ so that $K = JT$. The $j$-th minibatch with $j \in \{1, \ldots, J\}$ is denoted by $(x_{j,t}, y_{j,t})_{1 \leqslant t \leqslant T}$. In order to compress the network, we propose to solve the following constrained optimization problem.

**Problem 1** We want to

$$\underset{(W,b)\in C}{\text{minimize}} \ g(W, b) \tag{7}$$

with

$$C = \big\{(W,b) \in \mathbb{R}^{N \times M} \times \mathbb{R}^N \mid (\forall j \in \{1, \ldots, J\}) \quad \sum_{t=1}^{T} d^2_{\partial f(y_{j,t})}(Wx_{j,t} + b - y_{j,t}) \leqslant T\eta\big\}, \tag{8}$$

where $g$ is a sparsity measure defined on $\mathbb{R}^{N \times M} \times \mathbb{R}^N$ and $\eta \in [0, +\infty[$ is some accuracy tolerance.

Since, for every $j \in \{1, \ldots, J\}$, the function $(W, b) \mapsto \sum_{t=1}^{T} d^2_{\partial f(y_{j,t})}(Wx_{j,t} + b - y_{j,t})$ is continuous and convex, $C$ is a closed and convex subset of $\mathbb{R}^{N \times M} \times \mathbb{R}^N$. In addition, this set is nonempty when there exist $\overline{W} \in \mathbb{R}^{N \times M}$ and $\overline{b} \in \mathbb{R}^N$ such that, for every $j \in \{1, \ldots, J\}$ and $t \in \{1, \ldots, T\}$, $d^2_{\partial f(y_{j,t})}(\overline{W}x_{j,t} + \overline{b} - y_{j,t}) = 0$. As we have seen in Section 3.1, this condition is satisfied when $(\overline{W}, \overline{b})$ are the parameters of the uncompressed layer. Often, the sparsity of the weight matrix is the determining factor whereas the bias vector represents a small number of parameters, so that we can make the following assumption.

**Assumption 2** For every $W \in \mathbb{R}^{N \times M}$ and $b \in \mathbb{R}^N$, $g(W, b) = h(W)$ where $h$ is a function from $\mathbb{R}^{N \times M}$ to $\mathbb{R} \cup \{+\infty\}$, which is lower-semicontinuous, convex, and coercive (i.e. $\lim_{\|W\|_{\mathrm{F}} \to +\infty} h(W) = +\infty$). In addition, there exists $(\overline{W}, \overline{b}) \in C$ such that $h(\overline{W}) < +\infty$ and there exists $(j^*, t^*) \in \{1, \ldots, J\} \times \{1, \ldots, T\}$ such that $y_{j^*, t^*}$ lies in the interior of the range of $R$.

Under this assumption, the existence of a solution to Problem 1 is guaranteed (see Appendix A). A standard choice for such a function is the $\ell_1$-norm of the matrix elements, $h = \| \cdot \|_1$, but other convex sparsity measures could also be easily incorporated within this framework, e.g. group sparsity measures. Another point worth being noticed is that constraints other than (8) could be imposed. For example, one could make the following alternative choice for the constraint set

$$C = \big\{(W,b) \in \mathbb{R}^{N \times M} \times \mathbb{R}^N \mid \underset{j \in \{1, \ldots, J\}, t \in \{1, \ldots, T\}}{\sup} d_{\partial f(y_{j,t})}(Wx_{j,t} + b - y_{j,t}) \leqslant \sqrt{\eta}\big\}. \tag{9}$$

Although the resulting optimization problem could be tackled by the same kind of algorithm as the one we will propose, Constraint (8) leads to a simpler implementation.

### 3.3 Optimization algorithm

A standard proximal method for solving Problem 1 is the Douglas-Rachford algorithm (Lions & Mercier, 1979; Combettes & Pesquet, 2007). This algorithm alternates between a proximal step aiming at sparsifying the weight matrix and a projection step allowing a given accuracy to be reached. Assume that a solution to Problem 1 exists. Then, this algorithm reads as shown on the top of the next page.

The Douglas-Rachford algorithm uses parameters $\gamma \in ]0, +\infty[$ and $(\lambda_n)_{n \in \mathbb{N}}$ in $]0, 2[$ such that $\sum_{n \in \mathbb{N}} \lambda_n (2 - \lambda_n) = +\infty$. Throughout this article, $\mathrm{proj}_S$ denotes the projection onto a nonempty closed convex set $S$. Under these conditions, the sequence $(W_n, b_n)_{n \in \mathbb{N}}$ generated by Algorithm 1 is guaranteed to converge to a solution to Problem 1 if there exists $(\overline{W}, \overline{b}) \in C$ such $\overline{W}$ is a point in the relative interior of the domain of $h$ Combettes & Pesquet (2007) (see illustrations in Appendix E).

---

**Algorithm 1:** Douglas-Rachford algorithm for network compression

---

**Initialize :** $\widehat{W}_0 \in \mathbb{R}^{N \times M}$ and $b_0 \in \mathbb{R}^N$
**for** $n = 0, 1, \ldots$ **do**
$\quad$ $W_n = \mathrm{prox}_{\gamma h}(\widehat{W}_n)$
$\quad$ $(\widetilde{W}_n, \widetilde{b}_n) = \mathrm{proj}_C(2W_n - \widehat{W}_n, b_n)$
$\quad$ $\widehat{W}_{n+1} = \widehat{W}_n + \lambda_n(\widetilde{W}_n - W_n)$
$\quad$ $b_{n+1} = b_n + \lambda_n(\widetilde{b}_n - b_n)$.

---

The proximity operator of function $\gamma h$ has a closed-form for standard choices of sparsity measures[1]. For example, when $h = \| \cdot \|_1$, this operator reduces to a soft-thresholding (with threshold value $\gamma$) of the input matrix elements. In turn, since the convex set $C$ has an intricate form, an explicit expression of $\mathrm{proj}_C$ does not exist. Finding an efficient method for computing this projection for large datasets thus constitutes the main challenge in the use of the above Douglas-Rachford strategy, which we will discuss in the next section.

### 3.4 COMPUTATION OF THE PROJECTION ONTO THE CONSTRAINT SET

For every mini-batch index $j \in \{1, \ldots, J\}$, let us define the following convex function:

$$(\forall (W, b) \in \mathbb{R}^{N \times M} \times \mathbb{R}^N) \quad c_j(W, b) = \sum_{t=1}^{T} d_{\partial f(y_{j,t})}^2 (W x_{j,t} + b - y_{j,t}) - T\eta. \qquad (10)$$

Note that, for every $j \in \{1, \ldots, J\}$, function $c_j$ is differentiable and its gradient at $(W, b) \in \mathbb{R}^{N \times M} \times \mathbb{R}^N$ is given by

$$\nabla c_j(W, b) = (\nabla_{\mathsf{W}} c_j(W, b), \nabla_{\mathsf{b}} c_j(W, b)), \qquad (11)$$

where

$$\nabla_{\mathsf{W}} c_j(W, b) = 2 \sum_{t=1}^{T} e_{j,t} x_{j,t}^\top, \qquad \nabla_{\mathsf{b}} c_j(W, b) = 2 \sum_{t=1}^{T} e_{j,t} \qquad (12)$$

with

$$(\forall t \in \{1, \ldots, T\}) \quad e_{j,t} = W x_{j,t} + b - y_{j,t} - \mathrm{proj}_{\partial f(y_{j,t})}(W x_{j,t} + b - y_{j,t}). \qquad (13)$$

A pair of weight/bias parameters belongs to $C$ if and only if it lies in the intersection of the 0-lower level sets of the functions $(c_j)_{1 \leqslant j \leqslant J}$. To compute the projection of some $(W, b) \in \mathbb{R}^{N \times M} \times \mathbb{R}^N$ onto this intersection, we use Algorithm 2 ($\| \cdot \|_{\mathrm{F}}$ denotes here the Frobenius norm).

This iterative algorithm has the advantage of proceeding in a minibatch manner. It allows us to choose the mini-batch index $j_n$ at iteration $n$ in a quasi-cyclic manner. The simplest rule is to activate each minibatch once within $J$ successive iterations of the algorithm so that they correspond to an epoch. The proposed algorithm belongs to the family of block-iterative outer approximation schemes for solving constrained quadratic problems, which was introduced in (Combettes, 2003). The convergence of the sequence $(W_n, b_n)_{n \in \mathbb{N}}$ generated by Algorithm 2 to $\mathrm{proj}_C(W, b)$ is thus guaranteed. One of the main features of the algorithm is that it does not require to perform any projection onto the 0-lower level sets of the functions $c_j$, which would be intractable due to their expressions. Instead, these projections are implicitly replaced by subgradient projections, which are much easier to compute in our context.

### 3.5 DEALING WITH VARIOUS NONLINEARITIES

For any choice of activation operator $R$, we have to calculate the projection onto $\partial f(y)$ for every vector $y$ satisfying Eq. (1). This projection is indeed required in the computation of the gradients of functions $(c_j)_{1 \leqslant j \leqslant J}$, as shown by Eq. (13). Two properties may facilitate this calculation. First, if $f$ is differentiable at $y$, then $\partial f(y)$ reduces to a singleton containing the gradient $\nabla f(y)$ of

---

[1]http://proximity-operator.net

---

**Algorithm 2:** Minibatch algorithm for computing $\mathrm{proj}_C(W, b)$

---

**Initialize:** $W_0 = W$ and $b_0 = b$

**for** $n = 0, 1, \ldots$ **do**

    Select a batch of index $j_n \in \{1, \ldots, J\}$

    **if** $c_{j_n}(W_n, b_n) > 0$ **then**

        Compute $\nabla_{\mathsf{W}} c_{j_n}(W_n, b_n)$ and $\nabla_{\mathsf{b}} c_{j_n}(W_n, b_n)$ by using Eq. (12) and Eq. (13)

        $\delta W_n = \frac{c_{j_n}(W_n, b_n) \, \nabla_{\mathsf{W}} c_{j_n}(W_n, b_n)}{\|\nabla_{\mathsf{W}} c_{j_n,n}(W_n, b_n)\|_{\mathrm{F}}^2 + \|\nabla_{\mathsf{b}} c_{j_n}(W_n, b_n)\|^2}$

        $\delta b_n = \frac{c_{j_n}(W_n, b_n) \, \nabla_{\mathsf{b}} c_{j_n}(W_n, b_n)}{\|\nabla_{\mathsf{W}} c_{j_n,n}(W_n, b_n)\|_{\mathrm{F}}^2 + \|\nabla_{\mathsf{b}} c_{j_n}(W_n, b_n)\|^2}$

        $\pi_n = \mathrm{tr}((W_0 - W_n)^\top \delta W_n) + (b_0 - b_n)^\top \delta b_n$

        $\mu_n = \|W_0 - W_n\|_{\mathrm{F}}^2 + \|b_0 - b_n\|^2$

        $\nu_n = \|\delta W_n\|_{\mathrm{F}}^2 + \|\delta b_n\|^2$

        $\zeta_n = \mu_n \nu_n - \pi_n^2$

        **if** $\zeta_n = 0$ *and* $\pi_n \geqslant 0$ **then**

            $W_{n+1} = W_n - \delta W_n$

            $b_{n+1} = b_n - \delta b_n$

        **else if** $\zeta_n > 0$ *and* $\pi_n \nu_n \geqslant \zeta_n$ **then**

            $W_{n+1} = W_0 - (1 + \frac{\pi_n}{\nu_n}) \delta W_n$

            $b_{n+1} = b_0 - (1 + \frac{\pi_n}{\nu_n}) \delta b_n$

        **else**

            $W_{n+1} = W_n + \frac{\nu_n}{\zeta_n}(\pi_n(W_0 - W_n) - \mu_n \delta W_n)$

            $b_{n+1} = b_n + \frac{\nu_n}{\zeta_n}(\pi_n(b_0 - b_n) - \mu_n \delta b_n)$

    **else**

        $W_{n+1} = W_n$

        $b_{n+1} = b_n$

---

$f$ at $y$, so that, for every $z \in \mathbb{R}^N$, $\mathrm{proj}_{\partial f(y)}(z) = \nabla f(y)$. Second, $R$ is often separable, i.e. consists of the application of a scalar activation function $\rho \colon \mathbb{R} \to \mathbb{R}$ to each component of its input argument. According to our assumptions, there thus exists a proper lower-semicontinuous convex function $\varphi$ from $\mathbb{R}$ to $\mathbb{R} \cup \{+\infty\}$ such that $\rho = \mathrm{prox}_\varphi$ and, for every $z = (\zeta^{(k)})_{1 \leqslant k \leqslant N} \in \mathbb{R}^N$, $f(z) = \sum_{k=1}^N \varphi(\zeta^{(k)})$. This implies that, for every $z = (\zeta^{(k)})_{1 \leqslant k \leqslant N} \in \mathbb{R}^N$, $\mathrm{proj}_{\partial f(y)}(z) = (\mathrm{proj}_{\partial \varphi(v^{(k)})}(\zeta^{(k)}))_{1 \leqslant k \leqslant N}$, where the components of $y$ are denoted by $(v^{(k)})_{1 \leqslant k \leqslant N}$. Based on these properties, a list of standard activation functions $\rho$ is given in Table 1, for which we provide the associated expressions of the projection onto $\partial \varphi$. The calculations are detailed in Appendix B.

An example of non-separable activation operator frequently employed in neural network architectures is the softmax operation defined as: $(\forall z = (\zeta^{(k)})_{1 \leqslant k \leqslant N} \in \mathbb{R}^N)\ R(z) = \left( \frac{\exp(\zeta^{(k)})}{\sum_{j=1}^N \exp(\zeta^{(j)})} \right)_{1 \leqslant k \leqslant N}$.

It is shown in Appendix C that, for every $y = (v^{(k)})_{1 \leqslant k \leqslant N}$ in the range of $R$,

$$(\forall z \in \mathbb{R}^N) \quad \mathrm{proj}_{\partial f(y)}(z) = Q(y) + \frac{\mathbf{1}^\top(z - Q(y))}{N} \mathbf{1}, \tag{14}$$

where $\mathbf{1} = [1, \ldots, 1]^\top \in \mathbb{R}^N$ and $Q(y) = (\ln v^{(k)} + 1 - v^{(k)})_{1 \leqslant k \leqslant N}$.

## 3.6 SIS ON MULTI-LAYERED NETWORKS

Algorithm 3 describes how we make use of SIS for a multi-layered neural network. We use a pretrained network and part of the training sequence to extract layer-wise input-output features. Then we apply SIS on each individual layer $l$ by passing $\eta$, layer parameters $(W^{(l)}, b^{(l)})$ and extracted input-output features $(Y^{(l-1)}, Y^{(l)})$ to Algorithm 1. The benefit of applying SIS to each layer independently is that we can apply SIS on all the layers of a network in parallel. This reduces the time required to process the whole network and compute resources are optimally utilized.

| Name | $\rho(\zeta)$ | $\text{proj}_{\partial\varphi(v)}(\zeta)$ |
|---|---|---|
| **Sigmoid** | $(1 + e^{-\zeta})^{-1} - \frac{1}{2}$ | $\ln(v + 1/2) - \ln(v - 1/2) - v$ |
| **Arctangent** | $(2/\pi)\arctan(\zeta)$ | $\tan(\pi v/2) - v$ |
| **ReLU** | $\max\{\zeta, 0\}$ | $\begin{cases} 0 & \text{if } v > 0 \text{ or } \zeta \geqslant 0 \\ \zeta & \text{otherwise} \end{cases}$ |
| **Leaky ReLU** | $\begin{cases} \zeta & \text{if } \zeta > 0 \\ \alpha\zeta & \text{otherwise} \end{cases}$ | $\begin{cases} 0 & \text{if } v > 0 \\ (1/\alpha - 1)v & \text{otherwise} \end{cases}$ |
| **Capped ReLU** | $\text{ReLU}_\alpha(\zeta) = \min\{\max\{\zeta, 0\}, \alpha\}$ | $\begin{cases} \zeta & \text{if } (v = 0 \text{ and } \zeta < 0) \\ & \text{ or } (v = \alpha \text{ and } \zeta > 0) \\ 0 & \text{otherwise} \end{cases}$ |
| **ELU** | $\begin{cases} \zeta & \text{if } \zeta \geqslant 0 \\ \alpha(\exp(\zeta) - 1) & \text{otherwise} \end{cases}$ | $\begin{cases} 0 & \text{if } v > 0 \\ \ln\left(\frac{v+\alpha}{\alpha}\right) - v & \text{otherwise} \end{cases}$ |
| **QuadReLU** | $\dfrac{(\zeta + \alpha)\text{ReLU}_{2\alpha}(\zeta + \alpha)}{4\alpha}$ | $\begin{cases} v & \text{if } v = 0 \text{ and } \zeta \leqslant -\alpha \\ -v + 2\sqrt{\alpha v} - \alpha & \text{if } v \in ]0, \alpha] \\ & \text{ or } (v = 0 \text{ and } \zeta > -\alpha) \\ v - \alpha & \text{otherwise} \end{cases}$ |

Table 1: Expression of $\text{proj}_{\partial\varphi(v)}(\zeta)$ for $\zeta \in \mathbb{R}$ and $v$ in the range of $\rho$, for standard activation functions $\rho$. $\alpha$ is a positive constant.

---

**Algorithm 3:** Parallel SIS for multi-layered network

---

**Input:** input sequence $X \in \mathbb{R}^{M \times K}$, compression parameter $\eta > 0$, weight matrices
$\quad\quad W^{(1)}, \ldots, W^{(L)}$, and bias vectors $b^{(1)}, \ldots, b^{(L)}$
$Y^{(0)} \leftarrow X$
**for** $l = 1, \ldots, L$ **do**
$\quad | \quad Y^{(l)} = R_\ell(W^{(l)\top}Y^{(l-1)} + b^{(l)})$
$\quad | \quad \widehat{W}^{(l)}, \widehat{b}^{(l)} \leftarrow \text{SIS}(\eta, W^{(l)}, b^{(l)}, Y^{(l)}, Y^{(l-1)})$
**Output:** $\widehat{W}^{(1)}, \ldots, \widehat{W}^{(L)}$ and $\widehat{b}^{(1)}, \ldots, \widehat{b}^{(L)}$

---

## 4 EXPERIMENTS

In this section, we conduct various experiments to validate the effectiveness of SIS in terms of the test accuracy vs. sparsity and inference time FLOPs vs. sparsity by comparing against RigL (Evci et al., 2020). We also include SNIP (Lee et al., 2019), GraSP (Wang et al., 2020), SynFlow (Tanaka et al., 2020), STR (Kusupati et al., 2020), and FORCE (Jorge et al., 2020). These methods start training from a sparse network and have some limitations when compared to methods that prune a pretrained network Blalock et al. (2020); Gale et al. (2019). For a fair comparison we also include LRR (Renda et al., 2020) which uses a pretrained network and multiple rounds of pruning and retraining by leveraging learning rate rewinding. The experimental setup is described in Appendix D.

### 4.1 MODERN CONVNETS ON CIFAR AND IMAGENET

We compare SIS with competitive baselines on CIFAR-10/100 for three different sparsity regimes 90%, 95%, 98%, and the results are listed in Table 2. It can be observed that LRR, RigL and SIS are able to maintain high accuracy with increasing sparsity. LRR performs better than both RigL and SIS for VGG19 on CIFAR-10 at 90% and 95% sparsity. When compared to SNIP, our method achieves impressive performance for VGG19 on CIFAR-100 (58.46 -> 71.17). In the case of ResNet50, SIS outperforms all the other methods for CIFAR-10/100 except for CIFAR-100 at 90%.

| Dataset | CIFAR-10 | | | CIFAR-100 | | |
|---|---|---|---|---|---|---|
| Pruning ratio | 90% | 95% | 98% | 90% | 95% | 98% |
| **VGG19** (Baseline) | 94.23 | - | - | 74.16 | - | - |
| SNIP (Lee et al., 2019) | 93.63 | 93.43 | 92.05 | 72.84 | 71.83 | 58.46 |
| GraSP (Wang et al., 2020) | 93.30 | 93.04 | 92.19 | 71.95 | 71.23 | 68.90 |
| SynFlow (Tanaka et al., 2020) | 93.35 | 93.45 | 92.24 | 71.77 | 71.72 | 70.94 |
| STR (Kusupati et al., 2020) | 93.73 | 93.27 | 92.21 | 71.93 | 71.14 | 69.89 |
| FORCE (Jorge et al., 2020) | 93.87 | 93.30 | 92.25 | 71.9 | 71.73 | 70.96 |
| LRR (Renda et al., 2020) | **94.03** | **93.53** | 91.73 | 72.12 | 71.36 | 70.39 |
| RigL (Evci et al., 2020) | 93.47 | 93.35 | 93.14 | 71.82 | 71.53 | 70.71 |
| SIS (Ours) | 93.99 | 93.31 | **93.16** | **72.06** | **71.85** | **71.17** |
| **ResNet50** (Baseline) | 94.62 | - | - | 77.39 | - | - |
| SNIP (Lee et al., 2019) | 92.65 | 90.86 | 87.21 | 73.14 | 69.25 | 58.43 |
| GraSP (Wang et al., 2020) | 92.47 | 91.32 | 88.77 | 73.28 | 70.29 | 62.12 |
| SynFlow (Tanaka et al., 2020) | 92.49 | 91.22 | 88.82 | 73.37 | 70.37 | 62.17 |
| STR (Kusupati et al., 2020) | 92.59 | 91.35 | 88.75 | 73.45 | 70.45 | 62.34 |
| FORCE (Jorge et al., 2020) | 92.56 | 91.46 | 88.88 | 73.54 | 70.37 | 62.39 |
| LRR (Renda et al., 2020) | 92.62 | 91.27 | 89.11 | **74.13** | 70.38 | 62.47 |
| RigL (Evci et al., 2020) | 92.55 | 91.42 | 89.03 | 73.77 | 70.49 | 62.33 |
| SIS (Ours) | **92.81** | **91.69** | **90.11** | 73.81 | **70.62** | **62.75** |

Table 2: Test accuracy of sparse VGG19 and ResNet50 on CIFAR-10 and CIFAR-100 datasets.

| Sparsity | 60% | | 80% | | 90% | | 96.5% | |
|---|---|---|---|---|---|---|---|---|
| | Top-1 Acc(%) | FLOPs | Top-1 Acc(%) | FLOPs | Top-1 Acc(%) | FLOPs | Top-1 Acc(%) | FLOPs |
| SNIP (2019) | 73.95 | 1.88G | 69.67 | 941M | 65.30 | 409M | 54.70 | 292M |
| GraSP (2020) | 74.02 | 1.63G | 72.06 | 786M | 68.14 | 470M | 51.31 | 290M |
| SynFlow (2020) | 74.13 | 1.61G | 72.18 | 776M | 68.26 | 465M | 60.14 | 288M |
| STR (2020) | 76.82 | 1.59G | 75.29 | 705M | 74.13 | 341M | 67.20 | 117M |
| FORCE (2020) | 73.71 | 1.39G | 70.96 | 685M | 69.78 | 455M | 59.00 | 276M |
| LRR (2020) | **77.10** | 2.04G | 76.61 | 918M | 75.92 | 584M | 72.13 | 371M |
| RigL (2020) | 76.50 | 1.79G | 75.10 | 920M | 73.00 | 515M | 72.70 | 257M |
| SIS (Ours) | 77.05 | **1.34G** | **76.96** | **647M** | **76.31** | **298M** | **73.11** | **101M** |

Table 3: Test Top-1 accuracy and inference FLOPs of sparse ResNet50 on ImageNet where baseline accuracy and inference FLOPs are 77.37% and 4.14G, respectively.

| Sparsity | 75% | | | 90% | | |
|---|---|---|---|---|---|---|
| | LRR | RigL | SIS (Ours) | LRR | RigL | SIS (Ours) |
| V1 (70.90) | 68.79 | 69.97 | **70.11** | 66.59 | 67.10 | **67.15** |
| FLOPs (569M) | 498M | 461M | **367M** | 401M | 331M | **284M** |
| V2 (71.88) | 68.83 | 69.60 | **69.83** | 64.17 | **65.23** | 65.11 |
| FLOPs (300M) | 267M | 211M | **182M** | 192M | 174M | **162M** |
| V3 (72.80) | 68.97 | 70.21 | **70.47** | 64.32 | 65.13 | **66.07** |
| FLOPs (226M) | 187M | 198M | **172M** | 185M | 167M | **151M** |

Table 4: Test accuracy and inference FLOPs of sparse MobileNet versions using RigL and SIS on ImageNet, baseline accuracy and inference FLOPs shown in brackets.

Due to its small size and controlled nature, CIFAR-10/100 may not appear sufficient to draw solid conclusions. We thus conduct further experiments on ImageNet using ResNet50 and MobileNets. Table 3 shows that, in the case of ResNet50, LRR performs marginally better than SIS at 60% sparsity.

At 80%, 90%, and 96.5% sparsity SIS outperforms all other methods. For all sparsity regimes, SIS achieves least inference FLOPs. RigL achieves these results in same training time as SIS with less training time FLOPs. This may be related to the fact that SIS can achieve better compression in the last layer before SoftMax. MobileNets are compact architectures designed specifically for resource-constrained devices. Table 4 shows results for RigL and SIS on MobileNets. We observe that SIS outperforms all MobileNet versions at 75% sparsity level. For a 90% sparsity level, SIS outperforms RigL for MobileNet V1 and V3 whereas, for MobileNetV2, RigL performs slightly better than SIS at 90% sparsity level. In all the cases, we can see that the resulting SIS sparse network uses fewer FLOPs than RigL. A possible explanation for this fact is that SIS leverages activation function properties during the sparsification process.

## 4.2 SEQUENTIAL TASKS

| **Network** | JASPER | | Transformer-XL | | N-BEATS | |
|---|---|---|---|---|---|---|
| | WER | FLOPs | PPL | FLOPs | SMAPE | FLOPs |
| Dense | 12.2 | 4.53G | 18.6 | 927.73G | 8.3 | 41.26M |
| SNIP (Lee et al., 2019) | 14.3 | 2.74G | 24.6 | 398.92G | 10.1 | 21.45M |
| LRR (Renda et al., 2020) | 13.7 | 2.61G | 23.1 | 339.21G | **9.3** | 14.47M |
| RigL (Evci et al., 2020) | 13.9 | 2.69G | 22.4 | 326.56G | 10.2 | 15.13M |
| SIS (Ours) | **13.1** | **2.34G** | **21.1** | **290.38G** | 9.7 | **14.21M** |

Table 5: Test accuracy and inference FLOPs of JASPER, Transformer-XL, and N-BEATS at 70% sparsity.

**Jasper on LibriSpeech.** Jasper is a speech recognition model that uses 1D convolutions. The trained network is a 333 million parameter model and has a word error rate (WER) of 12.2 on the test set. We apply SIS on this network and compare it with RigL and SNIP in terms of sparsity. Table 5 reports WER and inference FLOPs for all three methods. SIS marginally performs better than SIS on this task in terms of WER and FLOPs for 70% sparsity. The main advantage of our approach lies in the fact that we can use a single pre-trained Jasper network and achieve different sparsity level for different types of deployment scenarios with less computational resources than RigL.

**Transformer-XL on WikiText-103.** Transformer-XL is a language model with 246 million parameters. The trained network on WikiText-103 has a perplexity score (PPL) of 18.6. In Table 5, we see that SIS performs better than SNIP and RigL in terms of PPL and has 68% fewer inference FLOPs. This is due to the fact that large language models can be efficiently trained and then compressed easily, but training a sparse sub-network from scratch is hard (Li et al., 2020), as is the case with SNIP and RigL. SNIP uses one-shot pruning to obtain a random sparse sub-network, whereas RigL is able to change its structure during training, which allows it to perform better than SNIP.

**N-BEATS on M4.** N-BEATS is a very deep residual fully-connected network to perform forecasting in univariate time-series problems. It is a 14 million parameter network. The Symmetric Mean Absolute Percentage Error (SMAPE) of the dense network on the M4 dataset is 8.3%. We apply SIS on this network and compare its performance with respect to RigL and SIS. SIS performs better than both methods and results in 65% fewer inference FLOPs.

## 5 CONCLUSION

In this article, we have proposed a novel method for sparsifying neural networks. The compression problem for each layer has been recast as the minimization of a sparsity measure under accuracy constraints. This constrained optimization problem has been solved by means of advanced convex optimization tools. The resulting SIS algorithm is i) reliable in terms of iteration convergence guarantees, ii) applicable to a wide range of activation operators, and iii) able to deal with large datasets split into mini-batches. Our numerical tests demonstrate that the approach is not only appealing from a theoretical viewpoint but also practically efficient.

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

APPENDICES

## A  EXISTENCE OF A SOLUTION TO PROBLEM 1 UNDER ASSUMPTION 2

Under Assumption 2, Problem 1 is equivalent to

$$\underset{(W,b)\in C}{\text{minimize}} \ h(W) \tag{15}$$

with

$$C = \big\{(W,b) \in \mathbb{R}^{N\times M} \times \mathbb{R}^{N} \ \big| \ \underset{j\in\{1,\dots,J\}}{\max} c_j(W,b) \leqslant 0\big\}, \tag{16}$$

where the functions $(c_j)_{1\leqslant j\leqslant J}$ are defined in Eq. (10). These functions being convex, $\Phi = \max_{j\in\{1,\dots,J\}} c_j$ is convex (Bauschke & Combettes, 2019, Proposition 8.16). We deduce that $\Psi = \inf_{b\in\mathbb{R}^N} \Phi(\cdot,b)$ is also a convex function (Bauschke & Combettes, 2019, Proposition 8.35). Since $\Phi \geqslant -\eta T$, $\Psi$ is finite valued. It is thus continuous on $\mathbb{R}^{N\times M}$ (Bauschke & Combettes, 2019, Corollary 8.40). Let us now consider the problem:

$$\underset{W\in\text{lev}_{\leqslant 0}\Psi}{\text{minimize}} \ h(W) \tag{17}$$

where $\text{lev}_{\leqslant 0}\Psi$ is the 0-lower level set of $\Psi$ defined as

$$\text{lev}_{\leqslant 0}\Psi = \big\{W \in \mathbb{R}^{N\times M} \ \big| \ \Psi(W) \leqslant 0\big\}, \tag{18}$$

$\Psi$ being both convex and continuous, $\text{lev}_{\leqslant 0}\Psi$ is closed and convex. According to Assumption 2, there exists $(\overline{W},\overline{b}) \in \mathbb{R}^{N\times M} \times \mathbb{R}^N$ such that $h(\overline{W}) < +\infty$ and $\Phi(\overline{W},\overline{b}) \leqslant 0$, which implies that $\Psi(\overline{W}) \leqslant 0$. This shows that $\text{lev}_{\leqslant 0}\Psi$ has a nonempty intersection with the domain of $h$. By invoking now the coercivity property of $h$, the existence of a solution $\widehat{W}$ to Problem (17) is guaranteed by standard convex analysis results (Bauschke & Combettes, 2019, Theorem 11.10).

To show that $(\widehat{W},\widehat{b})$ is a solution to (15), it is sufficient to show that there exists $\widehat{b} \in \mathbb{R}^N$ such that $\Phi(\widehat{W},\widehat{b}) = \Psi(\widehat{W})$. This is equivalent to prove that there exists a solution $\widehat{b}$ to the problem:

$$\underset{b\in\mathbb{R}^N}{\text{minimize}} \ \Phi(\widehat{W},b). \tag{19}$$

We know that $\Phi(\widehat{W},\cdot)$ is a continuous function. In addition, we have assumed that there exists $(j^*,t^*) \in \{1,\dots,J\} \times \{1,\dots,T\}$ such that $y_{j^*,t^*}$ is an interior point of $R(\mathbb{R}^N)$, which is also equal to the domain of $\partial f$ and a subset of the domain of $f$. Since $f$ is continuous on the interior of its domain, $\partial f(y_{j^*,t^*})$ is bounded (Bauschke & Combettes, 2019, Proposition 16.17(ii)). Then $d_{\partial f(y_{j^*,t^*})}$ is coercive, hence $c_{j^*}(\widehat{W},\cdot)$ is coercive, and so is $\Phi(\widehat{W},\cdot) \geqslant c_{j^*}(\widehat{W},\cdot)$. The existence of $\widehat{b}$ thus follows from the Weierstrass theorem.

## B  RESULTS IN TABLE 1

The results are derived from the expression of the convex function $\varphi$ associated with each activation function $\rho$ (Combettes & Pesquet, 2020a, Section 2.1) (Combettes & Pesquet, 2020b, Section 3.2).

**Sigmoid**

$$(\forall\zeta \in \mathbb{R}) \ \varphi(\zeta) = \begin{cases} (\zeta + 1/2)\ln(\zeta + 1/2) + (1/2 - \zeta)\ln(1/2 - \zeta) - \dfrac{1}{2}(\zeta^2 + 1/4) \\ \qquad\qquad\qquad\qquad\qquad\qquad\qquad\quad \text{if } |\zeta| < 1/2 \\ -1/4 \qquad\qquad\qquad\qquad\qquad\qquad\quad \text{if } |\zeta| = 1/2 \\ +\infty \qquad\qquad\qquad\qquad\qquad\qquad\quad\ \text{if } |\zeta| > 1/2. \end{cases} \tag{20}$$

The range of the Sigmoid function is $]-1/2, 1/2[$ and the above function is differentiable on this interval and its derivative at every $\upsilon \in ]-1/2, 1/2[$ is

$$\varphi'(\upsilon) = \ln(\upsilon + 1/2) - \ln(\upsilon - 1/2) - \upsilon. \tag{21}$$

We deduce that, for every $\zeta \in \mathbb{R}$, $\text{proj}_{\partial\varphi(\upsilon)}(\zeta) = \varphi'(\upsilon)$.

**Arctangent**

$$(\forall \zeta \in \mathbb{R}) \quad \varphi(\zeta) = \begin{cases} -\frac{2}{\pi} \ln\left(\cos\left(\frac{\pi\zeta}{2}\right)\right) - \frac{1}{2}\zeta^2, & \text{if } |\zeta| < 1 \\ +\infty, & \text{if } |\zeta| \geqslant 1. \end{cases} \tag{22}$$

By proceeding for this function similarly to the Sigmoid function, we have, for every $v \in \rho(\mathbb{R}) = ]-1, 1[$,

$$(\forall \zeta \in \mathbb{R}) \ \ \text{proj}_{\partial\varphi(v)}(\zeta) = \varphi'(v) = \tan(\pi v/2) - v. \tag{23}$$

**ReLU**

$$(\forall \zeta \in \mathbb{R}) \quad \varphi(\zeta) = \begin{cases} 0 & \text{if } \zeta \geqslant 0 \\ +\infty & \text{otherwise.} \end{cases} \tag{24}$$

For every $v \in \rho(\mathbb{R}) = [0, +\infty[$, we have

$$\partial\varphi(v) = \begin{cases} \{0\} & \text{if } v > 0 \\ ]-\infty, 0] & \text{if } v = 0. \end{cases} \tag{25}$$

We deduce that

$$(\forall \zeta \in \mathbb{R}) \quad \text{proj}_{\partial\varphi(v)}(\zeta) = \begin{cases} 0 & \text{if } v > 0 \text{ or } \zeta \geqslant 0 \\ \zeta & \text{otherwise.} \end{cases} \tag{26}$$

**Leaky ReLU**

$$(\forall \zeta \in \mathbb{R}) \qquad \varphi(\zeta) = \begin{cases} 0, & \text{if } \zeta > 0 \\ (1/\alpha - 1)\zeta^2/2 & \text{if } \zeta \leqslant 0. \end{cases} \tag{27}$$

Since this function is differentiable on $\mathbb{R}$, for every $v \in \mathbb{R}$,

$$(\forall \zeta \in \mathbb{R}) \quad \text{proj}_{\partial\varphi(v)}(\zeta) = \varphi'(v) = \begin{cases} 0 & \text{if } v > 0 \\ (1/\alpha - 1)v & \text{otherwise.} \end{cases} \tag{28}$$

**Capped ReLU**

$$(\forall \zeta \in \mathbb{R}) \qquad \varphi(\zeta) = \begin{cases} 0 & \text{if } \zeta \in [0, \alpha] \\ +\infty & \text{otherwise.} \end{cases} \tag{29}$$

We have thus for every $v \in [0, \alpha]$,

$$\partial\varphi(v) = \begin{cases} \{0\} & \text{if } v \in ]0, \alpha[ \\ ]-\infty, 0] & \text{if } v = 0 \\ [0, +\infty[ & \text{if } v = \alpha. \end{cases} \tag{30}$$

This leads to

$$(\forall \zeta \in \mathbb{R}) \quad \text{proj}_{\partial\varphi(v)}(\zeta) = \begin{cases} \zeta & \text{if } (v = 0 \text{ and } \zeta < 0) \\ & \text{or } (v = \alpha \text{ and } \zeta > 0) \\ 0 & \text{otherwise.} \end{cases} \tag{31}$$

**ELU**

$$(\forall \zeta \in \mathbb{R}) \qquad \varphi(\zeta) = \begin{cases} 0 & \text{if } \zeta \geqslant 0; \\ (\zeta + \alpha) \ln\left(\frac{\zeta+\alpha}{\alpha}\right) - \zeta - \frac{\zeta^2}{2}, & \text{if } -\alpha < \zeta < 0 \\ \alpha - \frac{\alpha^2}{2}, & \text{if } \zeta = -\alpha \\ +\infty, & \text{if } \zeta < -\alpha. \end{cases} \tag{32}$$

This function being differentiable on $\rho(\mathbb{R}) = ]-\alpha, +\infty[$, we have for every $v \in ]-\alpha, +\infty[$,

$$(\forall \zeta \in \mathbb{R}) \quad \text{proj}_{\partial\varphi(v)}(\zeta) = \varphi'(v) = \begin{cases} 0 & \text{if } v > 0 \\ \ln\left(\frac{v+\alpha}{\alpha}\right) - v & \text{otherwise.} \end{cases} \tag{33}$$

**QuadReLU**   Unlike the previous ones, this function does not seem to have been investigated before. It can be seen as a surrogate to the hard swish activation function, which is not a proximal activation function. Let us define

$$(\forall \zeta \in \mathbb{R}) \qquad \varphi(\zeta) = \begin{cases} +\infty & \text{if } \zeta < 0 \\ -\frac{\zeta^2}{2} + \frac{4}{3}\sqrt{\alpha}\zeta^{3/2} - \alpha\zeta & \text{if } \zeta \in [0, \alpha] \\ \frac{\zeta^2}{2} - \alpha\zeta + \frac{\alpha^2}{3} & \text{if } \zeta > \alpha. \end{cases} \tag{34}$$

$\varphi$ is a lower-semicontinuous convex function whose subdifferential is

$$(\forall v \in [0, +\infty[) \quad \partial\varphi(v) = \begin{cases} ]-\infty, -\alpha] & \text{if } v = 0 \\ \{-v + 2\sqrt{\alpha v} - \alpha\} & \text{if } v \in ]0, \alpha] \\ \{v - \alpha\} & \text{if } v > \alpha. \end{cases} \tag{35}$$

From the definition of the proximity operator, for every $(v, \zeta) \in \mathbb{R}^2$, we have $v = \text{prox}_\varphi(\zeta)$ if and only if

$$\zeta \in v + \partial\varphi(v) \quad \Leftrightarrow \quad \begin{cases} \zeta \in ]-\infty, -\alpha] & \text{if } v = 0 \\ \zeta = 2\sqrt{\alpha v} - \alpha & \text{if } v \in ]0, \alpha] \\ \zeta = 2v - \alpha & \text{if } v > \alpha. \end{cases}$$

$$\Leftrightarrow \quad v = \begin{cases} 0 & \text{if } \zeta \in ]-\infty, -\alpha] \\ \frac{(\zeta+\alpha)^2}{4\alpha} & \text{if } \zeta \in ]-\alpha, \alpha] \\ \frac{\zeta+\alpha}{2} & \text{if } \zeta > \alpha. \end{cases} \tag{36}$$

This shows that $\text{prox}_\varphi(\zeta) = (4\alpha)^{-1}(\zeta + \alpha)\,\text{ReLU}_{2\alpha}(\zeta + \alpha)$. In addition, for every $v \in [0, +\infty[$, it follows from Eq. (35) that the projection onto $\partial f(v)$ is

$$(\forall \zeta \in \mathbb{R}) \quad \text{proj}_{\partial f(v)}(\zeta) = \begin{cases} v & \text{if } v = 0 \text{ and } \zeta \leqslant -\alpha \\ -v + 2\sqrt{\alpha v} - \alpha & \text{if } v \in ]0, \alpha] \\ & \text{or } (v = 0 \text{ and } \zeta > -\alpha) \\ v - \alpha & \text{if } v > \alpha. \end{cases} \tag{37}$$

## C   Softmax Activation

Let $C$ denote the closed hypercube $[0, 1]^N$, let $V$ be the vector hyperplane defined as

$$V = \Big\{ z = (\zeta^{(k)})_{1 \leqslant k \leqslant N} \in \mathbb{R}^N \,\Big|\, \sum_{k=1}^{N} \zeta^{(k)} = 0 \Big\}, \tag{38}$$

and let $A$ be the affine hyperplane defined as

$$A = \Big\{ z = (\zeta^{(k)})_{1 \leqslant k \leqslant N} \in \mathbb{R}^N \,\Big|\, \sum_{k=1}^{N} \zeta^{(k)} = 1 \Big\} = V + u, \tag{39}$$

where $u = [1, \ldots, 1]^\top = 1/N \in \mathbb{R}^N$. If $R$ is the Softmax activation operator, the convex function $f$ such that $\text{prox}_f = R$ is (Combettes & Pesquet, 2020a, Example 2.23):

$$(\forall z = (\zeta^{(k)})_{1 \leqslant k \leqslant N}) \quad f(z) = \begin{cases} \sum_{i=1}^{N} \varphi(\zeta^{(k)}) & \text{if } z \in C \cap A \\ +\infty & \text{otherwise,} \end{cases} \tag{40}$$

where

$$(\forall \zeta \in [0, +\infty[) \quad \varphi(\zeta) = \zeta \ln \zeta - \frac{\zeta^2}{2} \tag{41}$$

(with the convention $0 \ln 0 = 0$). The latter function can be extended on $\mathbb{R}$, say by a quadratic function on $]-\infty, 0[$, yielding a convex function $\tilde{\varphi}$ which is differentiable on $\mathbb{R}$. We have then

$$(\forall z = (\zeta^{(k)})_{1 \leqslant k \leqslant N} \in \mathbb{R}^N) \quad f(z) = \sum_{i=1}^{N} \tilde{\varphi}(\zeta^{(k)}) + \iota_{C \cap A}(z), \tag{42}$$

where $\iota_{C \cap A}$ denotes the indicator function of the intersection of $C$ and $A$ (equal to 0 on this set and $+\infty$ elsewhere). It then follows from standard subdifferential calculus rules that, for every $y = (v^{(k)})_{1 \leqslant k \leqslant N} \in \mathbb{R}^N$,

$$\partial f(y) = (\tilde{\varphi}'(v^{(k)}))_{1 \leqslant k \leqslant N} + N_C(y) + N_A(y), \tag{43}$$

where $\tilde{\varphi}'$ is the derivative of $\tilde{\varphi}$ and $N_D$ denotes the normal cone to a nonempty closed convex set $D$, which is defined as

$$N_D(y) = \left\{ t \in \mathbb{R}^N \mid (\forall z \in D) \langle t \mid z - y \rangle \leqslant 0 \right\}. \tag{44}$$

Thus $N_A(y) = N_V(y)$ is the orthogonal space $V^\perp$ of $V$.
Let us now assume that $y \in R(\mathbb{R}^N) \subset ]0, 1[^N$. Then, since $y$ is an interior point of $C$, $N_C(y) = \{0\}$. We then deduce from Eq. (43) that

$$\partial f(y) = Q(y) + V^\perp, \tag{45}$$

where $Q(y) = (\varphi'(v^{(k)}))_{1 \leqslant k \leqslant N} = (\ln v^{(k)} + 1 - v^{(k)})_{1 \leqslant k \leqslant N}$. It follows that, for every $z \in \mathbb{R}^N$,

$$\mathrm{proj}_{\partial f(y)}(z) = Q(y) + \mathrm{proj}_{V^\perp}(z - Q(y)). \tag{46}$$

By using the expression of the projection $\mathrm{proj}_V = \mathrm{Id} - \mathrm{proj}_{V^\perp}$ onto hyperplane $V$, we finally obtain

$$\mathrm{proj}_{\partial f(y)}(z) = Q(y) + \frac{1^\top(z - Q(y))}{N} 1. \tag{47}$$

## D  EXPERIMENTAL SETUP

PyTorch is employed to implement our method. We use and extend SNIP and RigL code available here[2], LRR[3], GraSP[4], SynFlow[5], STR[6], and FORCE[7]. In order to manage our experiments we use Polyaxon[8] on a Kubernetes[9] cluster and use five computing nodes with eight V100 GPUs each. Floating point operations per second (FLOPs) is calculated as equal to one multiply-add accumulator using the code[10].

SIS has the following parameters: number of iterations of Algorithm 1, number of iterations of Algorithm 2, step size parameter $\gamma$ in Algorithm 1, constraint bound parameter $\eta$ used to control the sparsity, and relaxation parameter $\lambda_n \equiv \lambda$ of Algorithm 1. In our experiments, the maximum numbers of iterations of Algorithm 1 and Algorithm 2 are set to 2000 and 1000, respectively. $\lambda$ is set to 1.5 and $\gamma$ is set to 0.1 for all the SIS experiments. $\eta$ value depends on the network and dataset. With few experiments, we search for a good $\eta$ value that gives suitable sparsity and accuracy.

**VGG19 and ResNet50 on CIFAR-10/100.** We train VGG19 on CIFAR-10 for 160 epochs with a batch size of 128, learning rate of 0.1 and weight decay of $5 \times 10^{-4}$ applied at epochs 81 and 122. A momentum of 0.9 is used with stochastic gradient descent (SGD). We make use of 1000 images per training class when using SIS. We fine-tune the identified sparse subnetwork for 10 epochs at a learning rate of $10^{-3}$. For CIFAR-100 we keep the same training hyperparameters as for CIFAR-10. When applying SIS to the dense network, we use 300 images per class from the training samples. We fine-tune the identified sparse subnetwork for 40 epochs on the training set with a learning rate of $10^{-3}$. ResNet50 employs the same hyperparameters as VGG19, except the weight decay that we set to $10^{-4}$. When applying SIS to train dense ResNet50, we use the same partial training set and the same hyperparameters during fine-tuning. In case of VGG19 for CIFAR-10 and CIFAR-100, we found that $\eta$ values in range $(1.5, 2)$ works best for sparsity range $(90\%, 98\%)$. In case of ResNet50, $\eta$ values in range $(1, 2)$ is used.

---

[2]https://github.com/google-research/rigl
[3]https://github.com/lottery-ticket/rewinding-iclr20-public/tree/master/vision/gpu-src/official
[4]https://github.com/alecwangcq/GraSP
[5]https://github.com/ganguli-lab/Synaptic-Flow
[6]https://github.com/RAIVNLab/STR
[7]https://github.com/naver/force
[8]https://github.com/polyaxon/polyaxon
[9]https://kubernetes.io/
[10]https://github.com/Lyken17/pytorch-OpCounter

**ResNet50 on ImageNet**   We use the weights of ResNet50 pre-trained on ImageNet available at PyTorch hub[11]. When applying SIS to the dense pre-trained network we use 20% samples per class from the training set. We fine-tune the identified sparse subnetwork for 40 epochs on the training set with a learning rate of $10^{-4}$. We use different $\eta$ values in range $(0.7, 1.5)$ for sparsity range $(60\%, 90\%)$. We found that $\eta = 2.3$ achieves 96.5% sparsity.

**MobileNets on ImageNet**   We use MobileNetV1 dense pre-trained model from here[12] and MobileNetV2 from PyTorch hub[13]. In case of MobileNetV3, we replace the hard swish activation function used in the original paper Howard et al. (2019) with our QuadReLU function (see the last row of Table 1). We use hyperparameters provided in the original paper to train MobileNetV3. When applying SIS to the dense pre-trained MobileNets, we use 20% samples per class from the training set. We fine-tune the identified sparse subnetwork for 30 epochs on the training set with a learning rate of $10^{-4}$. For MobileNets, we search $\eta$ values in range $(0.6, 1.75)$ for sparsity range $(75, 90)$.

**Jasper on LibriSpeech**   A $B$x$R$ Jasper network has $B$ blocks, each consisting of $R$ repeating sub-blocks. Each sub-block consists of 1D-Convolution, Batch Normalization, ReLU activation, and Dropout. The kernel size of convolutions increases with depth. The network has one convolution block at the beginning and three at the end. We train a network of 13 encoding blocks and one decoding block, having 54 1D-Convolution layers on the LibriSpeech dataset. The total number of parameters in our trained network is 333 million. Jasper network is trained on train-clean-100, train-clean-360, and train-other-500 splits of the LibriSpeech dataset (Panayotov et al., 2015). The training configuration can be found here[14]. We use train-clean-100 when using SIS. We fine-tune the identified sparse sub-network on the completed training set for ten epochs with a learning rate of $10^{-4}$. We use $\eta$ values in range $(0.6, 1.75)$ for sparsity range $(70, 90)$.

**Transformer-XL on WikiText-103**   We train the Transformer-XL network (Dai et al., 2019b) on the base version of WikiText-103 (Merity et al., 2017). We use the training configuration available here[15]. We use 10% of the training set articles when using SIS. We use $\eta$ values in range $(0.5, 0.75)$ for sparsity range $(40, 70)$.

**N-BEATS on M4**   We train the interpretable architecture network of N-BEATS on the M4 dataset. The trained network has six residual blocks. Each block consists of four fully-connected layers and two linear projection layers. With 24 fully-connected layers, this network has 14 million trainable parameters. To compare different methods, we only train a single network on a 48-hour window instead of 180 networks on different timescales. We use the training configuration available here[16]. The training set has 50K time-series samples. We use 10K training samples to generate a sparse sub-network using SIS. We use $\eta$ values in range $(0.75, 1.5)$ for sparsity range $(70, 90)$.

## E   EMPIRICAL CONVERGENCE ANALYSIS

We illustrate the convergence of our method on LeNet-FCN trained on MNIST. LeNet-FCN is a fully-connected network having four layers with 784-300-1000-300-10 nodes (two 300 nodes and one 1000 node hidden layers). Figure 1 shows the convergence of SIS when applied to dense LeNet-FCN. We observe that the convergence is smooth and SIS finds a global solution for the first (ReLU activated) and last (softmax) layer cases. This fact is in agreement with our theoretical claims. SIS attains a sparsity of 99.21% at an error of 1.86%. The trained dense network has an error of 1.65%. This result is obtained at $\eta = 2$.

---

[11] https://pytorch.org/hub/pytorch_vision_resnet/
[12] https://github.com/RAIVNLab/STR
[13] https://pytorch.org/hub/pytorch_vision_mobilenet_v2/
[14] https://github.com/NVIDIA/DeepLearningExamples/blob/master/PyTorch/SpeechRecognition/Jasper/configs/jasper10x5dr_sp_offline_specaugment.toml
[15] https://github.com/NVIDIA/DeepLearningExamples/blob/master/PyTorch/LanguageModeling/Transformer-XL/pytorch/wt103_base.yaml
[16] https://github.com/ElementAI/N-BEATS/blob/master/experiments/m4/interpretable.gin

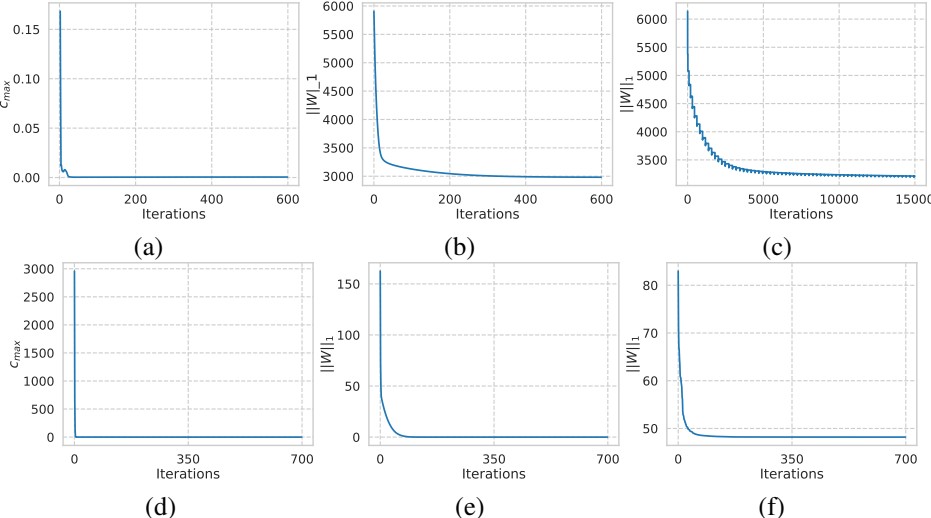

Figure 1: Convergence of SLIC: Top row shows the first layer (ReLU activated) and bottom row shows the last layer (softmaxed) in LeNet-FCN. (a) and (d) shows the evolution of the maximum value $c_{\max}$ of the constraint functions $(c_j)_{1 \leqslant j \leqslant J}$, (b) and (e) shows the evolution of $\|W\|_1$ in Algorithm 1 iterations. (c) and (f) shows $\|W\|_1$ evolution in Algorithm 2.

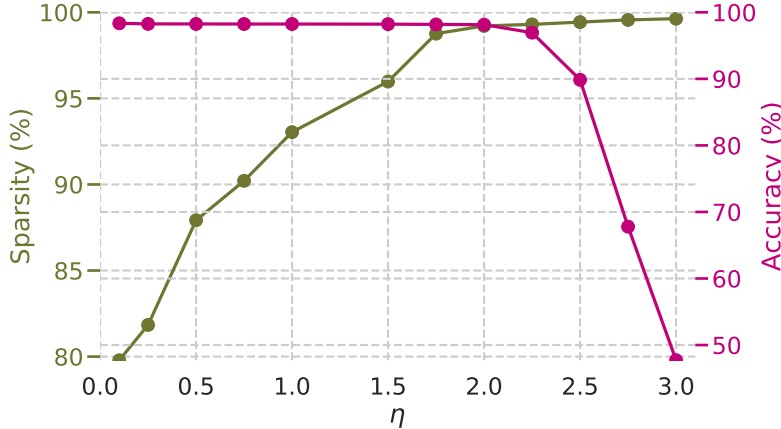

Figure 2: Effect of $\eta$ on LeNet-FCN

The $\eta$ parameter in our algorithm controls the accuracy tolerance. The higher, the more tolerant we are on the loss of precision and the sparser the network is. Thus, this parameter also controls the network sparsity. The choice of this parameter should be the result of an accuracy-sparsity trade-off. This is illustrated in Figure 2.

