# OpenReview forum: "Sparsifying Networks via Subdifferential Inclusion"
_ICLR.cc/2021/Conference — Reject_

### Official Review · AnonReviewer2 · 2020-10-26
**Interesting technique, but experiments need significant improvement**

**Rating:** 7
**Confidence:** 5

**Review:**

Summary:

The authors propose a new algorithm for inducing sparsity in the weights of neural networks after training. The proposed algorithm exploits the properties of commonly used activation functions to cast the sparsification problem as the minimization of a sparsity measure subject to approximation accuracy constraints. The proposed problem can be solved using convex optimization.

Pros:

The authors’ insights about popular activation functions and the approach used to cast the sparsification problem as a convex optimization problem are clever and interesting. The authors presented experiments on a wide range of deep learning models and tasks.

Cons:

Some of the details of the authors experiments are not clear or potentially misleading:
1. Some results presented for existing techniques are from those techniques’ original papers, but some results were re-run by the authors. For example, consider the ResNet-50 results on ImageNet (table 3, left). The RigL authors did not present results at 60% sparsity, and Appendix D does not include details on how this number was generated beyond the authors using the released code with RigL. The numbers at 80% and 90% sparsity are taken from the RigL paper. However, these numbers were achieved with 5x the number of training steps which was enabled by the reduced number of FLOPs used by RigL during training because it maintains a constant level of sparsity throughout the training process. They also use non-uniform distributions of sparsity across the layers of the network which affects the number of FLOPs in the resultant network. The author’s of this paper report a lower top-1 accuracy at 60% sparsity than the RigL paper reports at 80% sparsity, which leads me to believe that the training conditions (time, sparsity distribution) are not the same. Similarly, all of the RigL results for MobileNet family models (table 3, right) appear to have been run by the authors and the training setup details are not clear. For these results generated by the authors of this paper, they should also detail the amount of hyperparameter tuning performed for these baseline, as this can make a large difference in accuracy. I focused here on RigL because it appears to be the most commonly used baseline by the authors of this paper, but it seems likely that these observations apply to other techniques as well.
2. Using RigL as the “state-of-the-art” baseline for most comparisons is not entirely fair given it has additional capabilities (i.e., the ability to enable sparse training by maintaining a constant number of parameters across training) compared to the authors’ proposed post-training sparsification algorithm. Sparse training (i.e., sparse-to-sparse training) is known to be a more difficult problem than dense-to-sparse training [1] or post-training sparsification. It is good to include RigL for comparison, but this distinction should be made clear and other techniques that have comparable ability to the proposed technique should be included as well.

My suggestion to the authors are the following:
1. All results should be reported as accuracy with a given parameter count and accuracy with a given FLOP count. Ideally, these tradeoff curves should be plotted across a range of accuracies and FLOP counts. This helps to avoid many of the pitfalls in the comparisons of model compression approaches details by [2] and [3].
2. Make it clear that some algorithms under comparison have additional capabilities compared to the proposed approach (e.g., RigL with sparse training).
3. Add comparisons with other algorithms of similar capability to the proposed approach. The magnitude pruning approach of Zhu & Gupta [1] would be ideal for this I believe.

Comments:

The brackets are backwards in the last paragraph on page 4. I would encourage the authors to explain more of the background of their approach (proximal operators, convex optimization, etc.) in sections 3 and 4. Many of those working in model compression who would be interested in this work will not be familiar with these topics.

References:

1. https://arxiv.org/abs/1710.01878
2. https://arxiv.org/abs/1902.09574
3. https://arxiv.org/abs/2003.03033

[original score: 3 (clear rejection)]
11/24: Updated score based on updates from the authors. The addition of FLOP counts and more baselines in the experiments section greatly improved the paper. The proposed approach appears to achieve excellent FLOP-accuracy tradeoffs relative to existing approaches.

[2nd score: 6 (marginal acceptance)]
11/30: Updated score based on updates from the authors. The discrepancy with some baseline numbers has been resolved and the authors added clarifying information to the paper regarding the counting of FLOPs.

---

> ### Author Response · Authors · 2020-11-20
> **Response to Reviewer's comments**
>
> Thank you very much for taking time to review our paper. We greatly appreciate your criticisms and suggestions! Below, we address all concerns raised.
>
> *Some results presented for existing...*
>
> We ran experiments for which we did not find results in the literature. In case of RigL for 60\% and 96.5\% sparsity, we used the code provided by the authors as it is and specified the desired sparsity in the training script by setting the other hyperparameters as in the paper. We did the same for all other benchmarks. We believe that we could strengthen our experimental analysis by doing 3 runs with different seeds for all the methods, datasets and networks in order to draw more precise conclusions. We will report this in the final version of paper since it requires computation time.
>
> *Using RigL as the "state-of-the-art"...*
>
> We completely agree with reviewers point that RigL might not be fairly compared with SIS. Following the reviewer's comments, we have performed experiments with Learning Rate Rewinding method proposed by Renda et al., 2020. This approach has been recognized  to work best among existing methods that apply magnitude based pruning on pretrained networks. Please check the revised submission.
>
> *All results should be reported...*
>
> We have included FLOPs in Tables 2, 3, and 4.
>
> *Make it clear that some algorithms...*
>
> We have specified this in the first paragraph of Section 4 and also in Section 2.
>
> *Add comparisons with other other algorithms...*
>
> We have included the state-of-the-art magnitude based pruning method that works on pretrained networks, Learning Rate Rewinding (LRR) proposed by Renda et al., 2020.
> We would be happy to compare SIS with the interesting approach by Zhu and Gupta, but we were not able to find publicly available codes.
>
>
> *The brackets are backwards...*
>
> We use the European mathematical notation: $]a,b[$ is the open interval with lower bound $a$ and upper bound $b$
>
> *I would encourage the authors to explain...*
>
> The convex analysis tools are now introduced in a more comprehensive way in Section 3.1.
> Due to the lack of space, it is difficult to give more introductory materials. There however exist a number of nice tutorial papers about proximal methods (some of which we cite) and also the website http://proximity-operator.net gathers a lot of useful information.

---

> > ### Comment · AnonReviewer2 · 2020-11-24
> > **Response 1**
> >
> > Thank you to the authors for the updates. I think the addition of FLOP calculations & LRR as an additional baseline greatly improved the experimental results. I have a couple remaining concerns
> >
> > 1. The RigL results that you did not run (80%/90% sparsity in table 3) were run with 5x the number of training steps, but it looks like the experiments that you ran for 60% sparsity and 96.5% sparsity were run with less (I assume 1x). I think the best way to fix this oddity is for you to report their 1x training time results for 80%/90% sparsity in your table and then acknowledge in the text that RigL is a technique designed for sparse-to-sparse training, which could enable additional training time thanks to the training time FLOP savings.
> > 2. You should be clear that when you say "FLOPs" you mean FLOPs at inference time. This is clear in the introduction but some of the captions and text in the experiments section does not note this.
> > 3. You should report how you calculated FLOPs. They are often counted differently (e.g., with a multiply-add counted as 1) and it's easy to make mistakes in these calculations.
> >
> > The technique of Zhu & Gupta is implemented in the TensorFlow model pruning library (https://github.com/google-research/google-research/tree/master/model_pruning). I've increased my score based on the authors updates. I'm open to increasing it further if the authors address my above concerns.

---

> > > ### Author Response · Authors · 2020-11-25
> > > **Appreciating the quick response and further feedback**
> > >
> > > A warm thanks to the reviewer for giving time to the revised submission, increasing their score, and providing valuable feedback.
> > >
> > > *The RigL results that you...*
> > >
> > > Table  3  now  reflect  the  RigL  resuls  for  1x  training  time.   We  have  also mentioned this in the text.
> > >
> > > *Your should be clear...*
> > >
> > > We have changed all ”FLOPs” to ”inference FLOPs” in the text and table captions.
> > >
> > > *You should report how you...*
> > >
> > > Modern  CPUs  and  GPUs  support  fused  multiply-add  (FMA)  making  one multiplication and addition a single floating point operation (https://en.wikipedia.org/wiki/FMA_instruction_set). In our case 1 FLOP = 1 MAC, we have reported this in the Appendix D.
> > >
> > > Thanks for providing the working git link for Zhu & Gupta implementation.

---

### Official Review · AnonReviewer4 · 2020-10-28
**Highly impactful contribution with end-to-end results and analysis**

**Rating:** 9
**Confidence:** 3

**Review:**

## Summary
The authors pose sparsification as a subdifferential inclusion problem, a novel formulation that results in quite meaningful results on established benchmarks/tasks. The paper overall is very well-written with a detailed overview of current sparsification techniques and how the proposed method differs.

## Pros
* Very comprehensive analysis and proofs (which seem correct, although not thoroughly verified)
* Empirical results justify this novel approach across the board

## Suggestions
The computational characteristics of using SIS has not been characterized in the manuscript; it is no very clear what the complexity of training a large model is using the proposed approach. The authors suggest their training approach is efficient, but do not provide any empirical results or further justification. For example, all of the results in Table 3 and Table 4 can have an additional column that characterizes the time to train.

---

> ### Author Response · Authors · 2020-11-20
> **Response to Reviewer's comments**
>
> Thank you very much for your positive review. We greatly appreciate your comments and suggestions! Below, we address all concerns raised.
>
> *The computational characteristics...*
>
> In case of SIS and LRR, we use pretrained networks for compression. All other methods train a sparse network from scratch and take more epochs then training their dense counterparts. LRR on other hand goes thorough multiple rounds of pruning and fine-tuning to achieve desired sparsity. In case of SIS some time in form of algorithm 1 and 2 iterations is required to identify a sparse network and then few epochs for fine-tuning of sparse network. Since SIS, is applied in parallel on all layers of a network it is hard to compare different methods. We have however provided pruning and retraining details of SIS in Appendix D: Experimental Setup.

---

### Official Review · AnonReviewer1 · 2020-10-28
**The paper propose a compression scheme for NN that does not achieve SOTA but is an appealing competitors according the presented numerical experiments.**

**Rating:** 5
**Confidence:** 2

**Review:**

The paper propose a network compression algorithm by exploiting a reformulation of activation function as proximity operator. The latter is an optimization problem whose optimality condition reveals constraints on the weight matrix W of the neural net. The main idea is then to "biasedly" select W as a minimizer of a sparsity inducing penalties under a relaxation of the previous optimality conditions. The authors provide details on solving such problem as well as numerical experiments that leads to similar results than competitors.

- The proposed algorithm does not significantly improves the accuracy of estimators (eg convNet in Cifar) when compared with actual methods.

- The claim in the conclusion "SIS is reliable in term of convergence guarantee" is not supported by clear evidence. I did not find any such convergence proof in the paper. Once the SIS compression is used, it unclear that the same accuracy than the non-compressed NN is preserved.

- The core optimization problem (7) is solved approximately with Douglas-Rachford iterations. Neither the effect of the optimization error nor the selection of eta is clearly discussed.

- Should be nice if the authors can provide a pseudo-code of the overall SIS strategy in a practical deep neural net (not only one layer). As stated, the main idea is lost in the technical details for solving (7) (where there is few or no new contribution).

---

> ### Author Response · Authors · 2020-11-20
> **Response to Reviewer's comments**
>
> Thank you very much for taking time to review our paper. We greatly appreciate your comments and suggestions! Below, we address all concerns raised.
>
> *The proposed algorithm does not significantly improves...*
>
> Our method is based on principles different from existing methods, but it achieves comparable or better accuracies across various networks and datasets. In addition, in all  experiments, we observe that sparse networks generated by our method have best run time efficiency (least FLOPs). We think that this is is an evidence that our method better accounts the mathematical properties of activation function in the sparsification process.
>
> *The claim in the conclusion...*
>
> We provide an empirical convergence analysis in Appendix E. Theoretical convergence proofs for this type of optimization problem has been shown in the cited papers.
>
> *The core optimization problem...*
>
> Inded, the inner loop corresponding to the subgradient projection algorithm may introduce numerical errors because of the limited number of subiterations. However, first of all, the convergence of the Douglas-Rachford algorithm remains theoretically guaranteed if those errors are summable (Combettes & Pesquet, 2007).
> Secondly, we never observed any misbehavior in our practical experiments as we set the number of subiterations large enough. The choice of $\eta$ (one single parameter for each problem) was made empirically so as to reach the desired compression performance. We search the best value for $\eta$ by doing multiple experiments with the layer that has the highest number of parameters.
>
> *Should be nice if authors can provide...*
>
> We have provided the requested pseudo-code in Section 3.6 of the revised submission. We will release code, weights, and all experiment logs via our Polyaxon interface.

---

### Official Review · AnonReviewer3 · 2020-10-29
**In this paper, the authors propose a new model compression method based on subdifferential inclusion.**

**Rating:** 5
**Confidence:** 3

**Review:**

In this paper the authors propose a new model compression method based on subdifferential inclusion. The key idea is to make the outputs of the neurons in the sparse and dense networks at the same input close enough. They rewrite the activation function as the proximity operator of a proper convex function and finally formulate the compression problem into a constraint minimization problem using the technique of subdifferential inclusion. They conduct a series of experiments to evaluate the performance of their proposed methods.

Positive aspects:
1.	The idea of this paper makes sense.
2.	The experiment results show that the proposed method can achieve better performance than the baselines under this paper’s experimental setting.
3.	This paper is well written and easy to follow.

My concerns are:

1. In the proposed model, we need to choose proper value of $\eta$ for each layer, which is the required accuracy of the neuron’ output after compression. I understand that as reported by the authors in this paper, only in few experiments, they need to search for a good $\eta$. However, I think it is non-trivial to find proper value for $\eta$. I mean that different layers could have different tolerances on accuracy.  Since our goal is to achieve high test accuracy in the compressed network instead of the accuracy of the neuron’ output after compression, if we can find better values for $\eta$ we could achieve higher test accuracy. In other words, as it is challenging to find  near optimal $\eta$ for each layer, we could not reduce the network into very small size.

2. Because of my above concern, I recommend the authors to give more results of compressing networks into much smaller sizes. For example, in RigL, the size of ResNet50 on ImageNet is compressed by more than 97%.

---

> ### Author Response · Authors · 2020-11-20
> **Response to Reviewer's comments**
>
> Thank you very much for taking time to review our paper. We greatly appreciate your comments and suggestions! Below, we address all concerns raised.
>
> *In the proposed model, we need to choose proper value of each layer...*
>
> We agree with the reviewer that different layers may have different optimal values of $\eta$ for a target sparsity level. We first experimented with LeNet-FCN on MNIST to find how sensitive is the choice of $\eta$ at different layers to achieve a preset sparsity. What we found is that there is not much difference between $\eta$ values of any two different layers of a given network. In order to keep parameter tuning practical and to meet our compute budget, we did not search for optimal $\eta$ values for the all layers of this network according to our experiment. For a given network, we thus choose the $\eta$ value that we found best by experimenting with the layer that has the highest number of parameters and then use this $\eta$ to compress the whole network.
>
> *Because of my above concern, I recommend...*
>
> We have now included benchmarks and results for all methods applied on ResNet50 on ImageNet at 96.5\% sparsity. Please see Table 3 in the revised submission.

---

### Decision · Program_Chairs · 2021-01-07
**Final Decision**

**Decision:**

Reject

**Comment:**

I have serious concerns about how experiments are reported in this paper. Most methods tried to compare at an iteration complexity of roughly 100 epochs because it is known more computation improves performance very significantly but the computational resources are limited for many researchers, especially in academia. While this convention may not be the ideal way to compare different methods, for fairness, this practice has been followed in most of previous papers.

Unfortunately this paper disregarded this practice, and on Imagenet the reported results from previous works were mixed at 100 epochs (e.g. STR) and at 500 epochs (rigL — which was explicitly marked to be 5x in the original paper) without any clarification, and the only other method in the table showing comparable performance to the proposed method, LRR,  also requires many more than 100 epochs. Moreover, the authors did not explicitly disclose the equivalent epochs of their algorithms in the Imagenet experiments, and this is not acceptable. Based on the information inferred from the current writing, it is extremely likely that significant unfair advantages were given to the proposed algorithms.

Since the authors did not report experiments appropriately,  this paper cannot be accepted in its current form regardless of other potential merits of the proposed methods. I hope the authors view this outcome positively, and proactively fix the problem. If in revised versions, the experiments are reported according to the common practice, I am sure the work would become publishable.